# Provably Correct Automatic Subdifferentiation for Qualified Programs

**Sham M. Kakade**
University of Washington
sham@cs.washington.edu

**Jason D. Lee**
University of Southern California
jasonlee@marshall.usc.edu

## Abstract

The *Cheap Gradient Principle* [Griewank and Walther, 2008] — the computational cost of computing the gradient of a scalar-valued function is nearly the same (often within a factor of 5) as that of simply computing the function itself — is of central importance in optimization; it allows us to quickly obtain (high dimensional) gradients of scalar loss functions which are subsequently used in black box gradient-based optimization procedures. The current state of affairs is markedly different with regards to computing subderivatives: widely used ML libraries, including TensorFlow and PyTorch, do *not* correctly compute (generalized) subderivatives even on simple examples. This work considers the question: is there a *Cheap Subgradient Principle*? Our main result shows that, under certain restrictions on our library of nonsmooth functions (standard in nonlinear programming), provably correct generalized subderivatives can be computed at a computational cost that is within a (dimension-free) factor of 6 of the cost of computing the scalar function itself.

## 1 Introduction

The widespread implementation of Automatic Differentiation (AD) methods [Baydin et al., 2015] has had a transformative effect on applied machine learning; these methods have eased the difficulty for practitioners, across a range of disciplines, to learn sophisticated machine learning models (including deep neural architectures and richer inferential models). The paradigm is: one simply writes a program to compute the function of interest, say a scalar (loss) function $f(x) : \mathbb{R}^d \to \mathbb{R}$, and then a correctly implemented AD method will return both $f(x)$ and all $d$ of its partial derivatives when provided with $x$ as an input. These partial derivatives are often used in conjunction with some (stochastic) gradient-based optimization approach.

Underlying the effectiveness of this general black-box approach is the *Cheap Gradient Principle* [Griewank and Walther, 2008]: the computational cost of computing the vector of partial derivatives $(\partial f/\partial x_1, \partial f/\partial x_2, \ldots \partial f/\partial x_d)$ is often nearly the same as that of simply computing the scalar function $f(x)$ itself. In fact, for all rational functions, the striking Baur-Strassen theorem [Baur and Strassen, 1983, Griewank, 1989] shows that this increase in computational complexity is a (dimension free) factor of 5.

In many settings, our underlying function $f(x)$ is a nonsmooth function, and we resort to subgradient methods. This work considers the question: is there a *Cheap Subgradient Principle*? Specifically, given a program that computes a (locally Lipschitz) function $f$ and given a point $x$, can we automatically compute an element of the (Clarke) subdifferential $\partial f(x)$ [Clarke, 1975], and can we do this at a cost which is comparable to computing the function $f(x)$ itself? Informally, the set $\partial f(x)$ is the convex hull of limits of gradients at nearby differentiable points. It can be thought of as generalizing the gradient (for smooth functions) and the subgradient (for convex functions).

Let us briefly consider how current approaches handle nonsmooth functions, which are available to the user as functions in some library. Consider the following three equivalent ways to write the identity function, where $x \in \mathbb{R}$,

$$f_1(x) = x, \quad f_2(x) = \text{ReLU}(x) - \text{ReLU}(-x), \quad f_3(x) = 10 f_1(x) - 9 f_2(x),$$

where $\text{ReLU}(x) = \max\{x, 0\}$, and so $f_1(x) = f_2(x) = f_3(x)$. As these functions are differentiable at 0, the unique derivative is $f_1'(0) = f_2'(0) = f_3'(0) = 1$. However, both TensorFlow [Abadi et al., 2015] and PyTorch [Paszke et al., 2017], claim that $f_1'(0) = 1$, $f_2'(0) = 0$, $f_3'(0) = 10$. This particular answer is due to using a subgradient of 0 at $x = 0$. One may ask if a more judicious choice fixes such issues; unfortunately, it is not difficult to see that no such universal choice exists[1].

This example should be concerning for a number of reasons. The use of nonsmooth functions in AD go well beyond simple one dimensional nonsmooth functions (such as $\text{ReLU}(\cdot)$ or the $|\cdot|$); current methods permit utilizing eigenvalues, SVDs, QR decompositions (there are AD procedures on these nonsmooth linear algebra functions [Maclaurin et al., 2015, Seeger et al., 2017]).

**Is correctness important?** One option is to disregard these issues — which is the current state of affairs — based on the observation that in most cases these issues are unlikely to harm our optimization method. In numerical linear algebra, one could make the same argument: we never truly encounter degenerate linear systems (or degenerate eigenspaces); nonetheless, in retrospect, numerical issues have made evident the importance of carefully addressing these "corner cases". The situation may be analogous here: numerical issues in these approaches can easily lead to unstable outputs. Note that some numerical instability is certainly to be expected due to nonsmoothness (a point we return to in the Discussion under the notion of *mixed stability*); yet we would still hope to have nontrivial stability guarantees in our widely used AD libraries, much in the manner we have for our established numerical linear algebra libraries [Trefethen and Bau III, 1997, Demmel, 1997].

Ultimately, the importance of correctness in these methods is a decision that must be made by the broader ML community. Here, it is worthwhile to consider that AD software has a range of applications: from physical simulators to health care/social science applications to deployed online learning systems to differentiable programming. For example, when using physical simulators (say in robotics or in the sciences), a strong notion of stability may be critical when doing AD through nonsmooth system dynamics. In safety-critical settings, we may seek to have deployed online learning methods which are not susceptible to errors due to misspecified input-output behavior in our programs. Perhaps the most compelling reason for provably correct software implementations is to avoid costly failure modes due to the utilization of the methods in novel and unforeseen manners.

**Related Work:** These issues are in fact known in the mathematical AD literature (see Griewank and Walther [2008, Chapter 14]). Once we include either nonsmooth primitive functions or permit branching in a program, the usual chain rule fails to hold and incorrect input-out behavior is easy to observe. Due to that established calculus properties of nonsmooth functions [Klatte and Kummer, 2002, Mordukhovich, 2006] do not seem amenable to AD approaches, the current provable methods do not have general purpose, computationally efficient AD methods for subdifferentials.

One influential and powerful idea is that of lexicographic differentiation [Nesterov, 2005]; it is a property of a subclass of nonsmooth functions which allow these function to inherit a generalized notion of a chain rule. This idea has been utilized for obtaining correct generalized derivatives in Khan and Barton [2013], Griewank [2013]. The difficulty is that lexicographic approach often is expensive in that it involves a dimensional factor in the computational cost increase.

The other relatively few works that do focus on automatic generalized differentiation go through some notion of algorithmic linearization [A.Griewank, 1995, Nesterov, 2005, Khan and Barton, 2013, 2015, Fiege et al., 2017], where often piecewise smooth functions are considered, and the approach attempts at correct AD through probing the pieces through some linearization (see Griewank [2014] for review). The difficulties are due to understanding what information we can extract through linear "probes" into the function.

**Algorithm 1:** Straight Line Program for $f(x)$

---

**Input:** $x = (x_1, \ldots x_d)$
 1: **for** $k = d + 1, d + 2, \ldots T$ **do**
 2:     Compute:
$$x_k = g_k\big(x_{\text{parents}(k)}\big)$$
     where parents$(k)$ is the index set of the "parent" variables of $k$.
 3: **end for**
**Return:** $x_T$.

---

**Algorithm 2:** The Reverse Mode of AD

---

**Input:** variables $(x_1, \ldots x_T)$; a computational graph $\{\text{children}(t)\}_{t \in \{1, \ldots T\}}$; the associated
     derivatives
 1: Initialize: $\frac{\partial x_T}{\partial x_T} = 1$
 2: **for** $t = T, T - 1, \ldots 1$ **do**
 3:     Compute:
$$\frac{\partial x_T}{\partial x_t} = \sum_{i \in \text{children}(t)} \frac{\partial x_T}{\partial x_i} \frac{\partial x_i}{\partial x_t}$$
 4: **end for**
**Return:** $\frac{\partial x_T}{\partial x} = \left( \frac{\partial x_T}{\partial x_1}, \frac{\partial x_T}{\partial x_2}, \ldots \frac{\partial x_T}{\partial x_d} \right)$.

---

One of the first ideas along this line of thought is due to [A.Griewank, 1995], which shows how to compute directional derivatives of nonsmooth functions through following a "branch" the program would take on an input (where the branch corresponds to the approach direction in the directional derivative). In fact, our work uses this basic idea, as does the "branch locking" approach in Khan [2017], Griewank [2013]. The difficulty in these approaches is in finding a means to relate this linearization to properties of the (nonsmooth) functions, which will allow the algorithm to succeed; naively, we can tell when a method might have failed though it is difficult to guarantee if it will succeed.

As such, the extant body of work does not contain methods which contain only a constant factor blow up in the computational cost. Notable differences in this work is that our assumptions make strong connections to nonlinear programming [Abadie, 1967, Peterson, 1973, Gould and Tolle, 1971], which help in characterizing when the linearization approach is informative, and we provide a key technical result showing a certain chain rule holds for randomized algorithms. Furthermore, our focus is on generalizing the reverse mode for scalar functions (as opposed to focusing on multivariate functions where there is no known Cheap Gradient Principle).

**Our contributions:** Our main result provides — under a natural set of assumptions widely used in nonlinear programming — a provably correct Automatic Subdifferentiation procedure, which given some $x$, computes *both* the functional value $f(x)$ and a $d$ dimensional subdifferential $(u_1, \ldots u_d) \in \partial f(x)$, with a computational cost that is a factor of at most 6 times that of computing the scalar function $f(x)$ itself. Our assumption is that our library of functions be implemented in a manner consistent with the standard *constraint qualification* assumptions in nonlinear programming [Abadie, 1967]. In short, this work shows that in fact there is a *Cheap Subgradient Principle*.

## 2   Preliminaries

Assume $f \colon \mathbb{R}^d \to \mathbb{R}$ is a locally Lipschitz function, and recall, that by Rademacher's theorem, this implies that $f$ is differentiable almost everywhere. The *Clarke subdifferential* of $f$ at any point $x$ is the set [Clarke et al., 2008, Theorem 8.1]

$$\partial f(x) := \text{conv} \left\{ \lim_{i \to \infty} \nabla f(x_i) : x_i \xrightarrow{\Omega} x \right\}, \tag{1}$$

where $\Omega$ is any full-measure subset of $\mathbb{R}^d$ such that $f$ is differentiable at each of its points. Here, the limit is taken to be the set of all limit points. In classical circumstances, the subdifferential reduces to more familiar objects. Namely, when $f$ is $C^1$-smooth at $x$, the subdifferential $\partial f(x)$ consists only of the gradient $\nabla f(x)$, while for convex functions, it reduces to the subdifferential in the sense of convex analysis.

## 2.1 AD Review and The Baur-Strassen Theorem

A *straight line program* for computing $f(x) : \mathbb{R}^d \to \mathbb{R}$ is specified by a program of the form shown in Algorithm 1. Here the functions $g_1, g_2, \ldots$ are assumed to be some function from a library of functions. In the algebraic circuit complexity model, these functions are either monomials or affine functions of its inputs.

More generally, we will be interested in utilizing a richer class of functions where $g \in \mathcal{L}$, a library of functions, e.g. we may desire functions like the $|\cdot|$, ReLU $(x)$, or ever richer nonsmooth functions like eigenvalues.

Define $\mathrm{Runtime}(f; x)$ to be the time it takes to compute $f(x)$ under a given program for $f$.

**Theorem 2.1.** *[Baur and Strassen, 1983, Griewank, 1989] Assume all multiplications and additions have unit runtime cost. If we restrict to the algebraic circuit complexity model (where the functions $g_k$ are either monomials or affine functions), then it is possible to compute both $f(x)$ and all its partial derivatives $\nabla f(x)$ in time that is at most $5 * \mathrm{Runtime}(f; x)$.*

An algorithm achieving this guarantee is to first compute $f(x)$ and then use the *reverse mode* of AD, in Algorithm 2. To see the specific counting argument, see [Morgenstern, 1985]. This theorem is often more general: the reverse mode also correctly returns the derivatives even with a richer family of smooth functions in our library $\mathcal{L}$, often with a constant factor cost increase as well [Griewank, 1989]. The reverse mode itself has been rediscovered many times [Griewank, 2012]; the well known back-propagation algorithm [Rumelhart et al., 1986] is one example of the reverse mode of AD. The reverse mode (and the back-propagation algorithm) is not a direct application of the chain rule; the direct application of the chain rule is referred to as the *forward mode* of AD (see Griewank and Walther [2008]), which is $d$ times more expensive procedure to compute the gradient. The reverse mode can be viewed as a form of dynamic programming. To compare the two, in the reverse mode of AD, we compute the derivatives $\frac{\partial x_T}{\partial x_t}$, referred to as the adjoints[2], while in the forward mode of AD we would compute ($d$-dimensional) derivatives of the form $\frac{\partial x_t}{\partial x}$ (referred to as dual numbers).

## 2.2 Nonsmooth functions and our computational model

To specify how our nonsmooth functions are implemented, we extend the computational model to allow for branching, using (a restricted version[3] of) the Blum-Shub-Smale model of computation [Blum et al., 1988].

**Definition 2.1** (Computation Model). *The computational model for computing any $g(x) : \mathbb{R}^d \to \mathbb{R}$ in our library ($d$ may be different for each function) is specified by a program of the form shown in Algorithm 3. We assume that the function $g_{k,z}$ is either a monomial or an affine function of its inputs. Furthermore, for every $g$, we assume that there exists a time $T$, where the program terminates in at most this amount of time.*

Throughout, we make the following assumption:

**Assumption 2.1.** *(Computational Cost) Assume all multiplications and additions have unit runtime cost and that an execution of an "If" statement is also unit cost. For example, the cost of computing a monomial is the number of multiplications.*

The program implicitly encodes a function that has the following representation:

$$f(x) = \sum_{z \in \{-1,1\}^T} \mathbb{I}_{S_z}(x) p_z(x), \tag{2}$$

**Algorithm 3:** Program for a Nonsmooth function $g(x)$

---

**Input:** $x = (x_1, \ldots x_d)$
1: Initialize a vector $z$ to be all $-1$'s. $z$ is for notational convenience to keep track of the branch.
2: **for** $k = d + 1, d + 2, \ldots T$ **do**
3:   Compute:
$$x_k = g_{k,z}(x_{\mathrm{parents}(k,z)})$$

4:   If the program branches at $(k, z)$, then
      - If: $x_k \geqslant 0$, $z_k = 1$.
      - Else: $z_k = -1$.
5:   If the program halts at $(k, z)$, then terminate the **for** loop.
6: **end for**
**Return:** $x_k$.

---

| **Algorithm 4:** ReLU $(x)$ | **Algorithm 5:** ReLU $(x)$ |
|---|---|
| **Input:** $x = x_1$ | **Input:** $x = x_1$ |
| 1: Branch: | 1: Branch: |
|     • If: $x_1 \geqslant 0$, set $x_2 = x_1$. |     • If: $x_1^3 \geqslant 0$, set $x_2 = x_1$. |
|     • Else: set $x_2 = 0$. |     • Else: set $x_2 = 0$. |
| **Return:** $x_2$. | **Return:** $x_2$. |

Figure 1: Two programs that implement ReLU $(x)$: Both programs are correct and return the same value. However, the program on the right violates Assumption 3.1 since the gradient of the constraint function at $x = 0$, $\nabla(x_1^3) = 3x_1^2 = 0$.

where each $p_z$ is a polynomial; $\mathbb{I}_{S_z}$ is the indicator function on the set $S_z$; and $S_z$ consists of all $x$ where the program executes branch $z$ when given $x$ as input. The set $S_z$ can be explicitly defined as follows: for steps $k$ where the programs branches on $z$, define $h_{k,z}(x) = x_k$; on non-branching $k$, define $h_{k,z}(x) = -1$; define the vector valued function $h_z(x) = (h_1(z), \ldots h_T(x))$;

$$S_z := \{x | \, \mathbb{I}(\mathrm{sign}(h_z(x)) = z)\} \tag{3}$$

where the $\mathrm{sign}(\cdot)$ is the usual sign function (applied componentwise) taking values in $\{-1, 1\}$ (where we take $\mathrm{sign}(0) = 1$). Note that $S_z$ is specified by a set of polynomial inequalities as defined by the functions $h_{k,z}(x)$.

## 3 Provable Automatic Subdifferentiation

In the algebraic circuit complexity model, where AD is provably correct, branching is not permitted. The inclusion of branching into our program leads to a number of subtle issues. Branching allows us to implement the same nonsmooth function in different manners, which have important consequences in linearization approaches. Consider two different programs (with the same input-output behavior) for the ReLU $(x)$ function in Figure 1. The left program returns $x$ on the constraint set that is encoded as $S_1 = \{x | x \geqslant 0\}$, while the right program returns $x$ on the constraint set that is encoded as $S_1 = \{x | x^3 \geqslant 0\}$. In nonlinear programming, the importance of avoiding encoding constraints in the latter manner is well-known [Abadie, 1967, Peterson, 1973, Gould and Tolle, 1971].

This example motivates our restriction to only consider library functions that are encoded like the former set. We will make the standard constraint qualification assumption[4]. Roughly speaking, the assumption states that first order information characterizes the set of feasible perturbations. We state this assumption in a manner more directly applicable to our setting (see [Abadie, 1967, Peterson, 1973, Gould and Tolle, 1971]).

**Assumption 3.1.** *(Constraint Qualification on our Library) Assume for all $g \in \mathcal{L}$ that $g$ is locally Lipschitz and our program for $g$ (in our computational model) satisfies the constraint qualification condition on all sets $S_z$ in the following sense: suppose $\{h_z\}$ (for binary $z$) are the corresponding constraint functions in our program. For any $x, v$ (of the same input dimensionality of $g$), assume that for all $z$:*

$$\lim_{\delta \downarrow 0}(\mathrm{sign}(h_z(x + \delta v))) = \lim_{\delta \downarrow 0}(\mathrm{sign}(h_z(x) + \delta \nabla h_z(x) \cdot v)).$$

*Roughly, this states that the set approached along the limiting direction $x + \delta v$, when $\delta \downarrow 0$, can be determined with first order information.*

Before we state our main theorem, one more definition is in order, due to that $\mathrm{Runtime}(f; x)$ may not be continuous. Define the limiting runtime $\mathrm{Runtime}^*(f; x)$ of $f$ at $x$ as the (supremum) runtime to compute $f(x)$, as $x$ is approached from nearby points. Precisely,

$$\mathrm{Runtime}^*(f; x) := \sup\left\{\lim_{i \to \infty} \mathrm{Runtime}(f; x_i) : x_i \to x\right\},$$

(where the limit is taken to be the set of all limit points).

**Theorem 3.1.** *(A Cheap Subgradient Principle) Assume that our program for $f(x)$, in Algorithm 1, is allowed to use nonsmooth functions from our library $\mathcal{L}$ (in addition to affine functions and monomials). Suppose assumptions 2.1 and 3.1 hold. There exists a (randomized) algorithm, which upon input $x$, terminates in time that is at most $6 * \mathrm{Runtime}^*(f; x)$, and, almost surely, returns both $f(x)$ and an element $u \in \partial f(x)$.*

The following example shows one subtle issue with regards to constraint qualification.

**Example 3.1.** *(Constraint qualification on programs do not compose) Consider the function $f(x) = ReLU\left(x^2\right)$ (which is equivalent to the smooth function $x^2$). It is straight forward to see that the induced program for $f(x) = ReLU\left(x^2\right)$ (when we unravel it) does not satisfy the constraint qualification assumption, even if we do use an implementation of $ReLU(\cdot)$ that does satisfy this assumption. Regardless, in Example 3.4, we show that our algorithm does indeed correctly compute the gradient on this (continuous) function.*

Before we present the construction, we first provide a chain rule for nonsmooth functions.

## 3.1 A Chain Rule for Nonsmooth Functions

Let $D[g; v](x)$ denote the one-sided (Dini) directional derivative:

$$D[g; v](x) := \lim_{\delta \downarrow 0} \frac{g(x + \delta v) - g(x)}{\delta}.$$

(note that we are not assuming that $v$ is a unit vector). This derivative exists for all piecewise polynomials and semialgebraic functions [Coste, 2000, Lemma 6.2].

**Assumption 3.2.** *(Overloading the library with ASD subroutines) Assume we have a library of (locally Lipschitz) functions $\mathcal{L}$ computable in our computational model. For any $g \in \mathcal{L}$, with the representation $g(x) = \sum_{z \in \{-1,1\}^T} \mathbb{I}_{S_z}(x) p_z(x)$, assume we have the following associated automatic subdifferentiation subroutine $ASD[g]$ with the following behavior: upon input $(x; v)$, the output $[a, d, u] = ASD[g](x; v)$ satisfies*

$$a = g(x), \ d = D[g; v](x), \ u = \nabla p_z(x)$$

*where $z$ is such that:*

$$\lim_{\delta \downarrow 0}(\mathbb{I}_{S_z}(x + \delta v)) = 1.$$

*Roughly speaking, $u$ is the derivative determined by the set $S_z$ which is approached along the limiting direction $x + \delta v$, when $\delta \downarrow 0$.*

For any locally Lipschitz function $h$, define the limiting total derivate as:

$$\partial[h; v](x) := \lim_{\delta \downarrow 0} \nabla h(x + \delta v)$$

if the limit exists. For almost all $v$, the limit exists, and $\partial[h; v](x)$ is a subdifferential of $h$.

---

**Algorithm 6:** Automatic Subdifferentiation

---

**Input:** $x = (x_1, \ldots x_d)$, $v \in R^d$.
**Initialize:** Set $\dot{x}_1 = v_1$, $\dot{x}_2 = v_2, \ldots \dot{x}_d = v_d$.
  1: **for** $k = d+1, d+2, \ldots T$ **do**
  2:    Compute $[a, d, u] = \text{ASD}[g_k](x_{\text{parents}(k)}; \dot{x}_{\text{parents}(k)})$ and set:

$$x_k = a, \ \dot{x}_k = d, \quad \frac{\partial x_k}{\partial x_{\text{parents}(k)}} = u$$

  3: **end for**
  4: Compute $\frac{\partial x_T}{\partial x}$ using the Reverse Mode on these precomputed variables.
**Return:** $x_T$, and $\frac{\partial x_T}{\partial x}$.

---

**Theorem 3.2.** *(A Chain Rule for Nonsmooth Functions) Assume $h : \mathbb{R}^m \to \mathbb{R}$ and $g_1, \ldots g_m$ (where $g_i : \mathbb{R}^d \to \mathbb{R}$) are locally Lipschitz functions computable in our computational model and that the function $h$ is overloaded with an ASD subroutine as specified in Assumption 3.2. Define:*

$$f(x) := h(g_1(x), \ldots g_m(x)) = h(g(x)),$$

*where $g(x)$ is the vector valued function $(g_1(x), \ldots g_m(x))^\top$. Denote the $m \times 1$ vector of (one-sided) directional derivatives as $D[g; v](x)$. If it exists, let $\partial[g; v](x)$ denote $m \times d$ limiting Jacobian matrix (whose rows are given by the vectors $\partial[g_i; v](x)$'s). Set:*

$$[a, d, u] = ASD[h](g(x); D[g; v](x))$$

*For all but a measure $0$ set of $v$, we have that $\partial[f; v](x)$ and $\partial[g; v](x)$ exist and that:*

$$\partial[f; v](x) = \partial[g; v](x)^\top u. \tag{4}$$

**Example 3.2.** *Consider the example $x = f_2(x) = ReLU(x) - ReLU(-x)$. We define $h(y_1, y_2) = y_1 - y_2$, $g_1(x) = ReLU(x)$, and $g_2(x) = ReLU(-x)$, so that $f_2 = h(g_1(x), g_2(x))$. By applying the ASD subroutine to $h$, starting at $x = 0$ with $v = 1$ which leads to running $ASD[h]((0,0); (1,0)) = [a, d, u]$ (where it is straightforward to verify that $u = [1, -1]^\top$), we obtain*

$$\partial[f_2; v](0) = \partial[g; v](0)^T u$$
$$= \begin{bmatrix} 1 \\ 0 \end{bmatrix}^\top \begin{bmatrix} 1 \\ -1 \end{bmatrix}$$
$$= 1,$$

*which is correct. Furthermore, note a correct answer is obtained for any $v \neq 0$.*

**Example 3.3.** *We return to $f(x) = ReLU(x^2)$ from Example 3.1. Define $h(y) = ReLU(y)$, $g(x) = x^2$, and so $f(x) = h(g(x))$. By applying the chain rule lemma at $x = 0$ with $v = 1$,*

$$\partial[f; v](0) = \partial[g; v](0)u = 0 \cdot u = 0$$

*Subtly, note that $[a, d, u] = ASD[h](0; 0)$, so we are feeding a degenerate direction $d = 0$ into our subroutine. Regardless, the chain rule lemma still applies (for any $v$ in this case).*

## 3.2 The algorithm

We first present the algorithm that utilizes an overloaded library. We then provide a provably correct construction of this overloaded library. All proofs are provided in the appendix.

### Subdifferentiation with the overloaded library

Algorithm 6 is the Automatic Subdifferentiation procedure. Correctness follows from Lemma 3.1.

**Lemma 3.1.** *Suppose Assumptions 2.1 and 3.2 hold. Upon input of an arbitrary $x$, and if $v$ is sampled uniformly at random from the unit sphere, then, almost surely, Algorithm 6 returns both $f(x)$ and an element $u \in \partial f(x)$.*

**Algorithm 7:** Overloading the function $g(x)$

---

**Input:** $x = (x_1, \ldots x_d)$, $v \in R^d$.
**Initialize:** Set $\dot{x}_1 = v_1$, $\dot{x}_2 = v_2$, $\ldots \dot{x}_d = v_d$.
1: **for** $k = d + 1, d + 2, \ldots T$ **do**
2:      Compute $x_k$, its partial derivatives, and the directional derivative:

$$x_k = g_{k,z}(x_{\text{parents}(k,z)}), \quad \left\{ \frac{\partial x_k}{\partial x_j} \bigg| j \in \text{parents}(k,z) \right\},$$

$$\dot{x}_k = \sum_{j \in \text{parents}(k,z)} \frac{\partial x_k}{\partial x_j} \dot{x}_j$$

3:      If the program branches at $(k, z)$, then:
         •   If: $x_k > 0$, then $z_k = 1$.
         •   Elseif: $x_k = 0$ *and* $\dot{x}_k \geqslant 0$, then $z_k = 1$.
         •   Else: $z_k = -1$
4:      If the program halts at $(k, z)$, then terminate the **for** loop.
5: **end for**
6: Compute $\frac{\partial x_k}{\partial x}$ using the Reverse Mode on these pre-computed variables.
**Return:** $[a, d, u] = [x_k, \dot{x}_k, \frac{\partial x_k}{\partial x}]$.

---

*Proof.* Fix $k \in [d+1, \ldots, T]$. Every parent variable $j \in \text{parent}(k)$ can be expressed as $x_j = \tilde{g}_j(x)$, where $g_j$ is a piecewise polynomial on the $d$ dimensional input $x$. Thus

$$x_k = g_k(\tilde{g}_1(x), \ldots, \tilde{g}_{k-1}(x)).$$

Now the usual chain rule holds for directional derivatives [Shapiro, 1990]. As the forward mode of AD implements the usual chain rule of directional derivatives, then we have $\dot{x}_j = D[\tilde{g}_j; v]$.

By Assumption 3.2 and Theorem 3.2, $\text{ASD}[g_k](x_{\text{parents}(k)}, \dot{x}_{\text{parents}(k)})$ returns $u = \frac{\partial x_k}{\partial x_{\text{parents}(k)}} = \partial[g_k; \dot{x}_{\text{parents}(k)}]$ and this limiting total derivate satisfies the chain rule $\partial[x_k; v](x) = \partial[\tilde{g}; v](x)^\top u$. Since the limiting total derivates satisfies the chain rule and the validity of reverse mode AD algorithm relies only on the chain rule, Algorithm 6 correctly computes $\partial[f(x); v]$.

By Rademacher's theorem and the definition of Clarke subgradient in Equation (1), $\partial[f(x); v] \in \partial f(x)$, for almost all $v$.       $\square$

### Overloading the Library Functions

The following lemma shows that we can provide a method to correctly overload the library, which we use in Algorithm 6.

**Lemma 3.2.** *(Correct Library Overloading) Assume $g$ satisfies the constraint qualification conditions in Assumption 3.1, Suppose the corresponding representation is $g(x) = \sum_{z \in \{-1,1\}^T} \mathbb{I}_{S_z}(x) p_z(x)$, On an arbitrary input $x$ and $v$, Algorithm 7 returns $g(x)$, $D[g; v](x)$, and an element $u = \nabla p_z(x)$ where $z$ is such that: $\lim_{\delta \downarrow 0}(\mathbb{I}_{S_z}(x + \delta v)) = 1$.*

**Example 3.4.** *We again return to ReLU$(x^2)$ from Example 3.1. Here we examine how $h$ is overloaded based on the implementation in Algorithm 7. When $(x, v) = (0, 1)$, we are running $\text{ASD}[h](0; 0)$ and this may not follow the same branch had we run on the (infinitesimal) input $x = \epsilon v$ which leads to running $h(\epsilon^2 v^2)$. However, the gradient is correctly computed, $\partial \text{ReLU}(x^2) = 0$, regardless of the branch taken.*

## 4   Discussion and Open Questions

**Overloading the Library Functions:** It is not difficult to see that piecewise univariate functions can be implemented in our library.

**Algorithm 8:** $\sigma(x)$

---

**Input:** $x = x_1$
  1: Branch:
  - If: $x_1 \leqslant b_1$, set $x_2 = p_1(x)$.
  - Elseif: $x_1 \leqslant b_2$, set $x_2 = p_2(x)$.

    $\vdots$

  - Elseif: $x_1 \leqslant b_{k-1}$, set $x_2 = p_{k-1}(x)$.
  - Else: set $x_2 = p_k(x)$.

**Return:** $x_2$.

---

**Example 4.1.** *Univariate Piecewise Polynomial (Algorithm 8). Let $\sigma : \mathbb{R} \to \mathbb{R}$ be a univariate piecewise polynomial, meaning that the domain $\mathbb{R}$ is partitioned into a set of $k$ intervals $(-\infty, b_1), (b_1, b_2), \ldots, (b_{k-1}, \infty)$. On each interval, the function is equal to a polynomial $p_1, \ldots, p_k$.*

*Algorithm 8 provides a constraint qualified program for the function $\sigma(\cdot)$, which can be used as a library function.*

An important step would be in extending our computational model to allow the incorporation of provably correct automatic subdifferentiation libraries for linear algebra libraries. Auto-Grad [Maclaurin et al., 2015] does do AD through linear algebra methods though it can not be used to obtain correct subdifferentials in programs (at nondifferentiable points); obtaining correct generalized derivatives may be particularly important in cases where we deal with low rank methods. We conjecture our results can be extended, by extending the computational model, to handle these cases (there is already much known about the first order structure of these methods [Seeger et al., 2017]); technically, SVDs are not exactly computable in either the algebraic circuit complexity model or the Blum-Shub-Smale model.

**Numerical Analysis:** The most important open question is how to obtain numerically stable and accurate solutions [Trefethen and Bau III, 1997, Demmel, 1997]. We conjecture the techniques developed here will help in characterizing these issues. In particular, the most natural question is how to develop algorithms that satisfy the *mixed stability* criterion: the algorithm should give "nearly the right answer to nearly the right problem" (as in [Trefethen and Bau III, 1997]). For example, for the abs$(\cdot)$ function, it should be acceptable for an AD method to provide a subgradient near to $-1$ for a small input $\epsilon > 0$ due to roundoff error; however, it would undesirable for numerical error to lead vastly different gradients than those that arise from any nearby problem. This may be particularly important when doing AD in physical simulators.

**Acknowledgments:** We thank Dima Drusvyatskiy for many helpful discussions. Sham Kakade acknowledges funding from Washington Research Foundation Fund for Innovation in Data-Intensive Discovery, the NSF through award CCF-1740551, and ONR award N00014-18-1-2247. Jason D. Lee acknowledges support of the ARO under MURI Award W911NF-11-1-0303. This is part of the collaboration between US DOD, UK MOD and UK Engineering and Physical Research Council (EPSRC) under the Multidisciplinary University Research Initiative.

## Footnotes

[1] By defining $\text{ReLU}'(0) = 1/2$, the reader may note we obtain the correct derivative on $f_2, f_3$; however, consider $f_4(x) = \text{ReLU}(\text{ReLU}(x)) - \text{ReLU}(-x)$, which also equals $f_1(x)$. Here, we would need $\text{ReLU}'(0) = \frac{\sqrt{5}-1}{2}$ to obtain the correct answer.

[2]For a variable $x_T = g(x_{\mathrm{parents}})$, the notation $\frac{\partial x_T}{\partial x_t}$ refers to the derivative with respect to $x_t$, but holding all parent variables of $x_t$ as fixed. If $x_t$ is an input variable, then this is the usual partial derivative.

[3]We avoid halting concerns by assuming our programs halt in a bounded amount of time. We also explicitly avoid discussing tapes and registers in our computational cost model.

[4]The standard constraint qualification assumption on a constraint set is that the tangent cone of the constraint set equals the linearized cone (of the functions which define the constraints).

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
