[Reviews · NeurIPS 2018]

Reviewer 1



The paper addressed the problem of correctly computing sub-derivatives for nonsmooth scalar functions. And proved that this computation can be done efficiently under certain assumptions about the functions. In the introduction the importance for the problem was mentioned, however I am still not sure that the work is important in practice as the paper contains only theoretical results. Would be interesting to see if the new techniques improves the optimisation results in any experiments. Minor typos: - line 26: ...is often the nearly... - line 146: a function a function --------------- After reading the authors response and other reviews I agree that the problem is important and the paper is a good contribution to the theory. However, I still feel that the paper needs some experiments or numerical tests about runtime as suggested by reviewer 3. It is a pity that authors did not say anything about their experience for implementing and running their algorithms.

Reviewer 2



In this submission, the authors consider the problem of computing sub-differentiation for a class of non-smooth functions automatically and correctly. They give a very nice example that illustrates problems with current automated differentiation frameworks, such as tensorflow and pytorch. Then, the authors prove a chain rule for the one-sided directional derivative of a composite non-smooth function satisfying certain assumptions. Based on this rule, the authors derive a (randomized) algorithm for computing such a derivative for a particular kind of programs only with constant overhead. The algorithm is very similar to the one for back-ward automatic differentiation except that its forward computation is based on the newly-proved chain rule in the submission, rather than the standard chain rule for differentiation. I vote for the acceptance of the submission. Although the message in the submission is unlikely to change the practice of machine-learning engineers, it will at least make them be aware of potential pitfalls arising from differentiating ReLU or other non-smooth functions using tensorflow and pytorch. In the past, I annoyed my colleagues multiple times with the question: why is it ok to differentiate a non-smooth function like ReLU as in tensorflow or pytorch? I got a wide range of answers, such as justification based on sub-differentiation and argument based on measure-zero non-differentiability, but all of them were slightly ad-hoc. This submission seems to give a principled answer and to expose a line of past research and new results on this question. I learned a lot by reading the paper. I feel that other participants in NIPS would experience the same. * I think that it is better to give the definition of Clark sub-differential of f in the paper or in the appendix. * Assumption 3.1: How flexible is this requirement? * Don't Theorem 3.2 and Corollary 3.1 need Assumption 3.1? * Why do you have to sample v1,...,vd in Algorithm 6 uniformly in the sphere? If not, what would go wrong? * line 60: software is has ===> software has * line 101: it computes ===> computes * line 120: L is used here without definition. Say that it is a set of library functions. * line 146: a function a function ===> a function * line 181: allowed use ===> allowed to use * formula right after line 187: -g(v) ===> -g(x) * formula right after line 214: partial[f_2;v] ===> partial[f;v] * line 218: algorithm, that ===> algorithm that * I found it difficult to follow Example 3.3.

Reviewer 3



This paper aims to develop a cheap sub-differential principle for nonsmooth functions. They provide a randomized algorithm and show that under certain restrictions on their library of non-smooth functions, the randomized algorithm can generate sub-derivatives at a computational cost that is within a constant factor of 6 of the cost of computing the scalar function itself. The paper is not within the reviewer's expertise though the reviewer is very familiar with optimization and gradient-based methods. So the comments are mostly from the perspective of an optimization person outside of the field of "automatic differentiation methods". Strength: + Developing a cheap sub-differential principle is important for achieving computational efficient gradient-based optimization methods for non-smooth functions. + It looks like the paper has a method that can handle it. Weakness: - The paper is very hard to read and understand for a person who is not working on "automatic differentiation methods". The organization of the paper and the mathematical writing of the paper are not reader-friendly. For instance, almost all assumptions read like theorems. This may be common in the field of "automatic differentiation methods". But it looks confusing to the reviewer. - Continuing with the assumptions: It is hard to know how practical the assumptions are. The method relies on "overloading the library with ASD subroutines" (assumption 3.2). How hard is it? - The randomized algorithm needs to sample uniformly at random from the unit sphere. What is the computational cost of this procedure? I think this step won't scale up freely with the dimension of x. Therefore, the contribution of the method is questionable. - The paper needs to have numerical tests that report real run time on popular-used non-smooth functions where x is preferred to be high dimension. Typo: Theorem 3.1: "is allowed use" to "is allowed to use"